# Trends in cognitive outcomes in middle-aged Americans across three birth cohorts

Rebecca T. Brown[1,2,3,4*], Xi Zuo[5], Corey T. McMillan[6], Shana D. Stites[1,7], Irma T. Elo[8], Kenneth M. Langa[9,10], Jason Karlawish[1,4,6,11], Dawei Xie[5]

1 Division of Geriatric Medicine, Perelman School of Medicine of the University of Pennsylvania, Philadelphia, Pennsylvania, United States of America, 2 Geriatrics and Extended Care Program, Corporal Michael J. Crescenz VA Medical Center, Philadelphia, Pennsylvania, United States of America, 3 Center for Health Evaluation Research and Promotion, Corporal Michael J. Crescenz VA Medical Center, Philadelphia, Pennsylvania, United States of America, 4 Leonard Davis Institute of Health Economics, University of Pennsylvania, Philadelphia, Pennsylvania, United States of America, 5 Department of Biostatistics, Epidemiology, and Informatics, Perelman School of Medicine, University of Pennsylvania, Philadelphia, Pennsylvania, United States of America, 6 Department of Neurology, University of Pennsylvania, Philadelphia, Pennsylvania, United States of America, 7 Department of Psychiatry, Perelman School of Medicine at the University of Pennsylvania, Philadelphia, Pennsylvania, United States of America, 8 Department of Sociology, University of Pennsylvania, Philadelphia, Pennsylvania, United States of America, 9 Department of Internal Medicine, University of Michigan, Ann Arbor, Michigan, United States of America, 10 Institute for Social Research, University of Michigan, Ann Arbor, Michigan, United States of America, 11 Department of Medical Ethics and Health Policy, University of Pennsylvania, Philadelphia, Pennsylvania, United States of America

* rebecca.brown@pennmedicine.upenn.edu

## Abstract

Middle age is a key life course period for targeting modifiable risk factors for late-life cognitive decline. Although the prevalence of chronic conditions that are risk factors for cognitive impairment has increased in middle-aged adults since the 1990s, little is known about corresponding trends in cognitive trajectories and incidence of cognitive impairment. We conducted a cohort study of 7,852 participants ages 50–56 enrolled from 1998–2010 in the Health and Retirement Study (HRS) without cognitive impairment at baseline. Participants were followed biennially to examine cognitive trajectories and new-onset cognitive impairment by age 65, based on the HRS cognitive test battery. We used mixed effects models to examine change in cognitive scores and Cox regression models to compare incidence of cognitive impairment no dementia (CIND) across three birth cohorts (1942–1947, "War Babies"; 1948–1953, "Early Baby Boomers"; 1954–1959, "Mid Baby Boomers"). Mid Boomers had lower baseline cognitive scores compared to earlier cohorts but a slower rate of cognitive decline. Hazards of CIND were higher among White Early and Mid Boomers compared to War Babies in the first half of follow-up, but lower in the second half. In both trajectory and incidence analyses, those with lower educational attainment and racial/ethnic minorities were at higher risk for worse cognitive outcomes. Findings show mixed trends in cognitive outcomes among middle-aged Americans. Overall, Mid Boomers had lower baseline cognitive scores but slower decline compared to earlier cohorts. However,

**Data availability statement:** Data are freely available at the Health and Retirement Study website (https://hrsdata.isr.umich.edu/data-products/rand).

**Funding:** This work was supported by the National Institute on Aging at the National Institutes of Health (grant number R01AG070885 to R.T.B.; grant numbers K23AG065442 and K23AG065442-03S1 to S.D.S.; P30AG072979 to C.T.M) and Penn Institute on Aging (no grant number to C.T.M.). The Health and Retirement Study is supported by the National Institute on Aging (grant number U01AG009740) and performed at the Institute for Social Research, University of Michigan. The funders had no role in study design, data collection and analysis, decision to publish, or preparation of the manuscript.

**Competing interests:** The authors have declared that no competing interests exist.

disparities in cognitive outcomes persisted among those with lower educational attainment and racial/ethnic minorities.

## Introduction

With growing evidence that Alzheimer's disease and Alzheimer's disease related dementias (AD/ADRD) develop over decades, middle age is increasingly recognized as a key life course period to target modifiable risk factors for cognitive decline and AD/ADRD [1–4]. Many mid-life characteristics are associated with an increased risk for cognitive impairment or dementia later in life, including chronic conditions such as hypertension and obesity and health-related behaviors such as excessive alcohol use [5–14]. The 2024 Lancet Commission estimates that addressing these mid-life risk factors could result in up to 30% of dementias being prevented or delayed [2]. At the same time, there is growing evidence that health status has worsened in middle-aged Americans since the 1990s compared to earlier birth cohorts, including increases in the prevalence of chronic medical conditions that are risk factors for AD/ADRD [15–18]. This trend suggests the need to look beyond individual-level risk factors to understand how one's birth cohort may influence risk for health outcomes, possibly through exposures in the broader social or physical environment. This would allow for population-focused efforts to improve health outcomes for this age group and potentially reduce overall dementia burden. Yet little is known about whether cognitive outcomes have worsened in middle-aged people over this time period, as have rates of chronic conditions.

While the causes of the recent declines in health status in middle-aged Americans are still unclear, several studies have shown that middle-aged adults with lower socioeconomic status (SES), including lower education, income, and net worth, are at highest risk for these health declines [15–18]. Moreover, SES disparities in prevalence of chronic conditions and mortality risk in the U.S. have worsened substantially over the past decades, a trend occurring in parallel with economic stagnation for lower-income households [19,20]. Based on these trends, researchers hypothesize that lower SES is leading to worsening health status in middle-aged adults in more recent birth cohorts through exposures in the social and physical environment [21–23]. A growing number of studies have linked such environmental factors to risk of cognitive decline and AD/ADRD in older adults [24–30], and these factors could also be impacting cognitive trajectories and risk of AD/ADRD in middle age. However, despite increases in the prevalence of AD/ADRD risk factors in more recent birth cohorts compared to earlier birth cohorts, little is known about trends in cognitive outcomes in middle-aged adults.

Several prior studies have examined trends in outcomes related to cognition; one study showed that from 1999 to 2017, mortality from neurodegenerative diseases increased 88% in people ages 55–64 years [31]. Several cross-sectional studies have also shown that the prevalence of difficulty performing cognitively complex instrumental activities of daily living (IADLs) remained stable or increased in middle-aged people over the past 2–3 decades [17,18,32]. Another study found that

the prevalence of cognitive limitations among individuals ages 55–69 was stable from 1998 to 2014, but showed substantial disparities by socioeconomic status [33]. Taken together, these studies suggest that the prevalence of cognitive impairment in middle age has remained stable or worsened over the past two decades, particularly among those with lower socioeconomic status. However, these studies assessed prevalent cognitive impairment. Thus, they do not distinguish between longstanding impairments due to congenital conditions, trauma, or early health events versus cognitive changes that develop in middle age and may have different clinical implications.

To examine if cognitive outcomes have worsened in more recent birth cohorts of middle-aged adults compared to those born earlier, we examined cognitive trajectories and incidence and hazards of cognitive decline in the form of "cognitive impairment no dementia" (CIND, often a precursor of dementia) in middle age among three birth cohorts: War Babies (born 1942–1947); Early Baby Boomers (born 1948–1953); and Mid Baby Boomers (born 1954–1959). We also examined how cognitive trajectories and hazards of CIND differ among subgroups including persons with lower socioeconomic status and racial/ethnic minorities.

## Methods

### Setting and participants

We conducted a cohort study using longitudinal data from the Health and Retirement Study (HRS). The HRS is an ongoing nationally representative panel survey examining changes in health and wealth among Americans over age 50 [34]. The first participants were enrolled in 1992, and additional participants are enrolled every 6 years, so the sample remains representative of the population over age 50. Participants are interviewed every 2 years by telephone, face-to-face, or web.

Our analytic sample included participants who were ages 50–56 on enrollment in the 1998, 2004, or 2010 HRS survey waves. We excluded participants enrolled in 1992 because different cognitive measures were used in the 1992 and 1994 waves of HRS. We also excluded participants enrolled in 2016 due to the limited follow-up waves currently available; while available data extend through 2020, we excluded 2020 data due to the impact of the COVID-19 pandemic on in-person data collection. For participants enrolled in 1998, 2004, and 2010, respectively, we used a consistent follow-up period (1998–2006, 2004–2012, and 2010–2018). Of 10,296 participants in these waves who were ages 50–56 at enrollment, we excluded 5 who were in a nursing home at baseline, 1,203 who had a zero-sampling weight, which is typically due to being a spouse younger than age 50 of an HRS participant, and 1,236 who had an HRS cognitive score consistent with CIND or dementia at enrollment. Our final analysis included longitudinal data from 7,852 participants, collected at approximately 2-year intervals through 2018 (Fig 1).

The Institutional Review Board of the University of Pennsylvania approved the study. Participants provide verbal or written informed consent for participation in the HRS; because we used deidentified HRS data for this study, participant consent was waived by the Institutional Review Board.

### Measures

**Outcomes.** The co-primary outcomes were cognitive trajectories and incidence of CIND before age 65 based on repeated self- or proxy-reported cognitive status measures. Cognitive status in HRS is measured biennially using the HRS cognitive test battery which includes selected items from the Telephone Interview for Cognitive Status (TICS) [35–39]. The TICS is a validated screening instrument modeled on the Mini-Mental State Examination and developed for use in population-based studies. During each biennial study interview, participants complete cognitive tests including the immediate and delayed free recall test with 10 nouns, the serial 7 subtraction test, and the backward count test from 20 to 0 (total possible score of 0–27; lower scores indicate more severe impairment) [38]. These are valid cognitive tests for use in mid-life and assess memory, working memory, attention, and processing speed (S1 Table) [39–41]. Individuals with a score of 0–6 are classified as having dementia, 7–11 as having CIND, and 12–27 as having normal cognition [38]. These

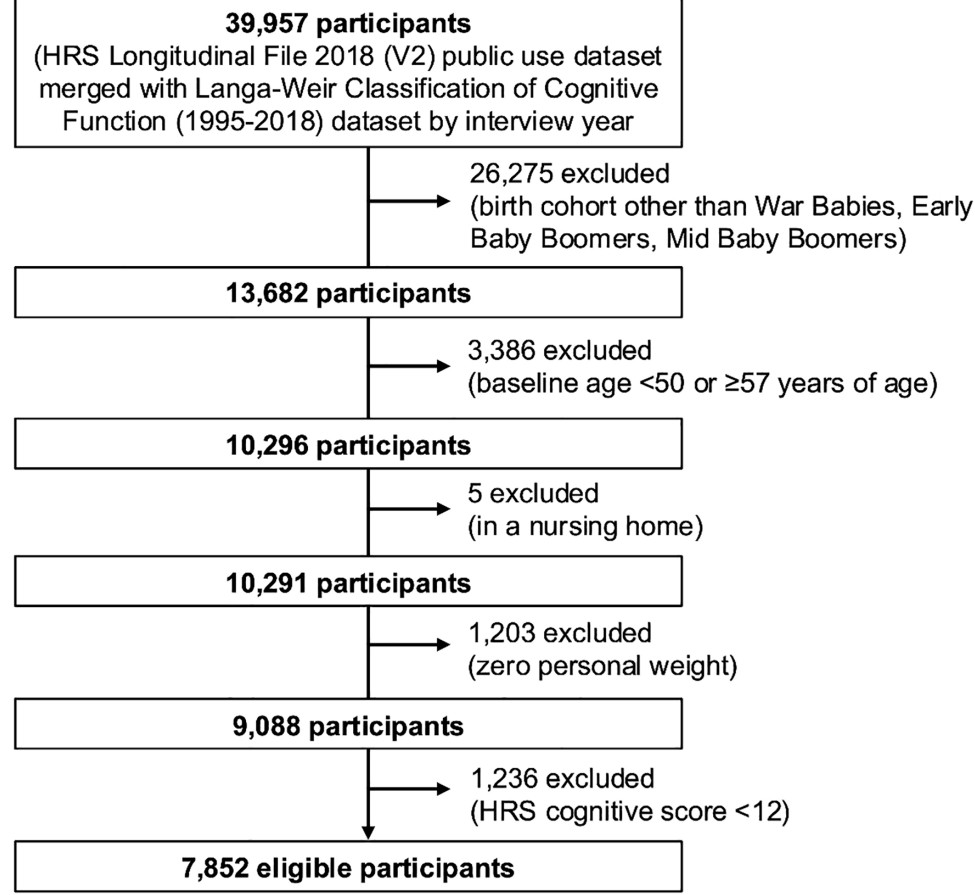

**Fig 1. Study Selection Flow Chart.** Note. HRS = Health and Retirement Study. V2 = Version 2. The figure shows the flow chart for selection of study participants.

cut-points were validated in the Aging, Demographics, and Memory Study (ADAMS), which used 3- to 4-hour in-home neuropsychological and clinical assessment and expert clinician adjudication to establish a gold-standard diagnosis of CIND or dementia [38,42].

Of note, this score threshold was originally validated for persons ages 70 and older. However, prior research has found that use of this cut-off to classify CIND status results in increasing prevalence of CIND across 5-year age bands (i.e., 55–59, 60–64, 65–69), suggesting good concurrent validity in younger age groups [33].

For participants represented by a proxy respondent enrolled in 2004 or 2010, an 11-point scale, also validated using ADAMS, was used to classify cognition [38]. The scale includes 3 measures: the proxy's rating of the participant's memory (0–4 scale: excellent, very good, good, fair, poor); the proxy's assessment of limitations in the participant's ability to perform 5 IADLs (preparing meals, shopping, managing medications, using the telephone, managing money; possible score of 0–5, with higher scores indicating more limitations); and the interviewer's assessment of whether the participant had difficulty completing the interview due to cognitive impairment (possible score of 0–2; no difficulty [0], some difficulty [1], or difficulty preventing completion [2]). Participants with a combined score of 6–11 are classified as having dementia, 3–5 as CIND, and 0–2 as normal cognition. For participants enrolled in 1998, only the first two proxy-based measures (memory rating and IADL limitations) were used, creating a combined 9-point scale. On this scale, participants with a score of 5–9

are classified as having dementia, 3–4 as CIND, and 0–2 as normal cognition. In a time-split validation study comparing the performance of this diagnostic algorithm to the ADAMS gold standard, the algorithm correctly classified 91% of participants in the training dataset and 92% in the validation dataset [43].

For proxy scores, we imputed an HRS cognitive score by randomly selecting a value from the corresponding cognitive category (normal cognition, CIND, or dementia). For instance, if a participant's 11-point proxy score fell between 3–5, classifying them as CIND, we would impute an HRS cognitive score within the range of 7–11.

We also examined incidence of persistent CIND, assessed as a time-to-event outcome. To do so, we first identified the initial episode of CIND or dementia based on self- or proxy-reported cognitive measures. To increase the likelihood of correct classification, we defined persistent CIND as two or more consecutive interviews indicating CIND or dementia [44], with time to persistent CIND calculated from baseline to first occurrence. We assumed a linear cognitive function change between the dates of interview before and after CIND or dementia and interpolated the actual CIND occurrence date.

**Main exposure and other measures.** The primary exposure was birth cohort. We included three birth cohorts in our analysis, enrolled in 1998, 2004, and 2010, respectively: War Babies (born 1942–1947); Early Baby Boomers (born 1948–1953); and Mid Baby Boomers (born 1954–1959).

We also considered several additional key variables based on prior literature and guided by the Lancet Commission life course model of dementia risk and the MacArthur Network on Socioeconomic Status and Health conceptual framework (Fig 2) [2,45–47]. The Lancet Commission model was developed to inform population-level studies of dementia risk and provides a framework for examining how risk and protective factors from different phases of the life course affect cognition. Because lower SES appears to play a role in worsening health status in middle-aged individuals in more recent birth cohorts, our conceptual model also draws on the work of the MacArthur Network on Socioeconomic Status and health, which has explored the role of SES in health disparities and how SES may influence health outcomes.[46,47] SES reflects an individual's economic and social position and is typically characterized by several measures, include education, income, and net worth. In their evidence-based model, the MacArthur Network proposes that the components of SES influence health outcomes via their effect on downstream mediators, including environmental resources and constraints, psychological influences, health behaviors, and access to health care. In turn, these factors lead to an increased risk of chronic conditions, which are known risk factors for CIND and dementia. The current study focuses on understanding how cognitive outcomes vary across birth cohorts, rather than on examining mechanisms of cognitive decline or identifying novel risk factors. For this reason, we focus on variables that are established risk factors for cognitive decline in this proposed causal pathway and have been extensively examined in prior studies examining trends in the prevalence of

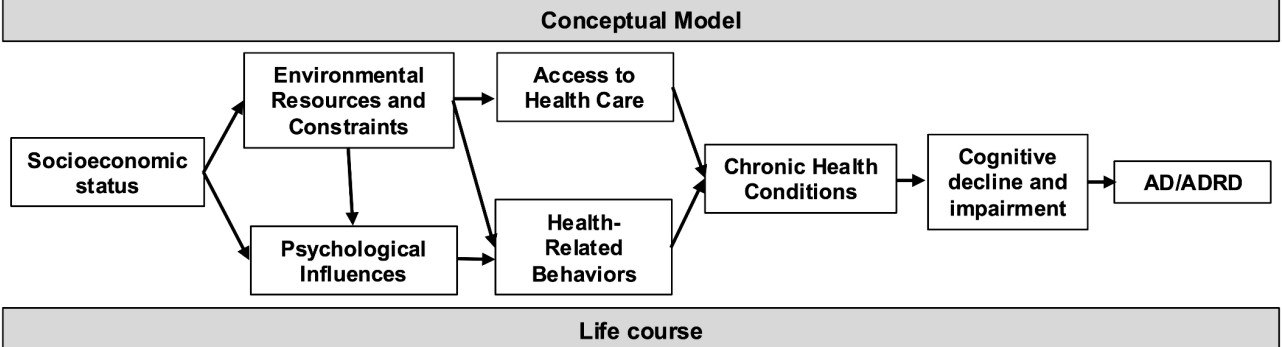

**Fig 2. Conceptual Model.** Note. AD/ADRD = Alzheimer's Disease and Alzheimer's Disease Related Dementias. The figure shows a conceptual model of the proposed causal pathway leading from scoioeconomic status to chronic health conditions and cognitive outcomes, which draws from the Lancet Commission life course model of dementia risk and the MacArthur Network on Socioeconomic Status and Health conceptual model.

cognitive impairment among older adults, including sociodemographic characteristics, measures of SES, and measures of health status.[2,45] Self-reported sociodemographic characteristics at study entry included age, gender, and race/ethnicity, and measures of SES included educational attainment and household net worth, calculated based on self-reported assets minus debts and adjusted for inflation using the Consumer Price Index for all Urban Consumers (CPI-U) [48]. Measures of health status included self-reported medical conditions (hypertension, stroke, diabetes, cardiac disease) and body mass index, calculated using self-reported weight and height.

## Statistical analysis

We first compared baseline characteristics across the three birth cohorts. P values were calculated using ANOVA for continuous variables and Chi-square tests for categorical variables. Two-sided p values were reported with p < .05 as statistical significance. These and the following analyses account for the complex HRS survey design including weight, cluster, and stratum.

We then used latent class growth models (group-based trajectory models) to identify distinct classes of trajectories of cognitive change. We used the Akaike information criterion (AIC) to determine the number of trajectory groups and the degree of each curve (e.g., linear, quadratic, cubic). We visually checked the linearity of the trajectories. We used multi-nominal logistic regression models to examine the association between birth cohort and other baseline characteristics with group membership. After confirming that the trajectories could be approximated by a straight line, we next fit linear mixed-effects models by assuming a random intercept and random slope for each participant with slope indicating the rate of cognitive decline. The intercepts and slopes were assumed to follow normal distributions. In linear mixed-effects models, time was modeled as years since baseline.

To compare the incidence of persistent CIND across the three birth cohorts, we used Kaplan-Meier curves. We accounted for the competing risk of death in two ways. First, we censored at the time of death to model cause-specific hazard in Cox regressions; second, we used Fine and Gray's subdistribution hazard model [49]. For the Cox regression models, the time axis was time since baseline, and participants were right censored by death or end of follow-up. For variables violating the proportional hazard assumption in the Cox regression models, we divided the total length of follow-up into two periods and estimated separate hazard ratios for each period. Participants were censored at age 65 or at their last interview before age 65. In sensitivity analyses for both the linear mixed-effects and Cox regression models, we excluded proxy respondents to assess the robustness of the cohort comparisons, given changes in HRS data collection procedures over time.

We conducted unadjusted and adjusted analyses for both the mixed effects and time-to-event analyses. We adjusted for covariates for both the intercept and slope for the mixed effects model. For all adjusted analyses, we considered the same set of covariates [45]. We also pre-specified several interactions with birth cohort, including education, race/ethnicity, and total net worth. The proportion of missing data in the variables was 0% with the exception of race/ethnicity (0.22%), educational attainment (0.04%), and BMI (1.49%). Due to the low percentage of missingness, we used complete case analysis in which we excluded those participants with missing data. Analyses were performed using SAS 9.4 (SAS Institute Inc., Cary, NC).

## Results

### Participant characteristics

Of the 7,852 participants, 24.5% (weighted) were War Babies, 31.6% Early Boomers, and 43.9% Mid Boomers. The baseline characteristics of participants varied across cohorts (Table 1). Compared to War Babies and Early Boomers, Mid Boomers were slightly older and more likely to identify as Black non-Latino, Latino, or of another race/ethnicity. Early Boomers had higher levels of educational attainment compared to both War Babies and Mid Boomers, with a greater percentage completing some college or a college degree. Mid Boomers had lower income and net worth compared to the

**Table 1. Baseline Characteristics of 7,852 Middle-Aged Participants Overall and by Birth Cohort.**

| Characteristics | Participants, weighted column % | | | |
| --- | --- | --- | --- | --- |
| | Overall (N = 7,852) | War Babies, born 1942–1947 (N = 1,926) | Early Baby Boomers, born 1948–1953 (N = 2,480) | Mid Baby Boomers, born 1954–1959 (N = 3,446) |
| Age, mean years (SD) | 53.0 (1.4) | 52.9 (1.2) | 53.0 (1.4) | 53.2 (1.5) |
| Gender, weighted % | | | | |
| Woman | 46.6 | 46.4 | 45.1 | 48.1 |
| Man | 53.4 | 53.6 | 54.9 | 51.9 |
| Race/ethnicity, weighted % | | | | |
| White non-Latino | 78.3 | 82.1 | 78.2 | 74.7 |
| Black non-Latino | 9.9 | 8.9 | 10.2 | 10.5 |
| Other | 4.1 | 2.7 | 4.3 | 5.2 |
| Latino | 7.8 | 6.3 | 7.3 | 9.6 |
| Educational attainment, weighted % | | | | |
| Less than high school | 13.0 | 14.8 | 11.0 | 13.1 |
| High school or GED | 25.6 | 30.2 | 23.1 | 23.5 |
| Some college | 28.9 | 25.8 | 31.0 | 29.9 |
| College or higher | 32.5 | 29.2 | 34.9 | 33.6 |
| Income, quartile, $, weighted % | | | | |
| ≤ $10,559 | 13.5 | 12.3 | 13 | 15.1 |
| > $10,559, ≤ $22,916 | 18.8 | 17.7 | 17 | 21.5 |
| > $22,916, ≤ $44,083 | 29.9 | 31.1 | 31.7 | 27.1 |
| > 44,083 | 37.8 | 38.9 | 38.3 | 36.3 |
| Net worth, quartile, $, weighted % | | | | |
| ≤ $17,527 | 24.2 | 19.8 | 21.4 | 30.9 |
| > $17,527, ≤ $71,657 | 25.1 | 28.1 | 22.8 | 24.4 |
| > $71,657, ≤ $188,325 | 24.6 | 27.3 | 25.2 | 21.5 |
| > $188,325 | 26.1 | 24.8 | 30.7 | 23.2 |
| Chronic medical conditions, weighted % | | | | |
| Hypertension | 32.2 | 28.2 | 30.6 | 37.5 |
| Stroke | 2 | 2.2 | 1.8 | 1.9 |
| Diabetes | 10.2 | 7.9 | 9.7 | 12.7 |
| Cardiac disease | 8.5 | 7.4 | 8.9 | 9.4 |
| Body mass index, weighted % | | | | |
| < 18.5 | 1.2 | 1.3 | 1.2 | 1.1 |
| 18.5-24.9 | 27.0 | 29.0 | 28.2 | 24.2 |
| 25-29.9 | 40.1 | 41.1 | 40.9 | 38.3 |
| ≥ 30 | 31.7 | 28.6 | 29.7 | 36.4 |
| HRS cognitive score with proxy score imputation, mean (SD) | 17.6 (2.5) | 18.0 (2.2) | 17.6 (2.4) | 17.3 (2.6) |
| Respondent type, weighted % | | | | |
| Self | 95.5 | 94.3 | 93.9 | 98.2 |
| Proxy | 4.5 | 5.7 | 6.1 | 1.8 |

*Note.* SE = standard error. GED = General Educational Development test. HRS = Health and Retirement Study. Analyses incorporate survey weights, strata, and clusters to account for the complex HRS survey design.

other cohorts. Rates of most chronic medical conditions were higher among Mid Boomers, including hypertension, diabetes, and obesity. Mid Boomers also had lower baseline HRS cognitive scores compared to the other two cohorts. A lower proportion of Mid Boomer respondents were proxies, which was consistent across follow-up waves (S2 Table).

### Longitudinal analyses of cognitive decline

**Group-based trajectory models.** The best group-based trajectory model, as identified by AIC values, revealed four distinct groups (Fig 3). The main distinction between the groups was in their baseline HRS cognitive scores, while trajectories of cognitive decline were approximately linear for all groups. Group 1 had the lowest baseline score, at 14.5, and Group 4 had the highest, at 21.5. The birth cohorts made up significantly different proportions of each trajectory group (S3 Table). War Babies comprised the largest proportion of Group 1 (34.5%) and Group 4 (39.8%), the groups with the lowest baseline cognitive score and the highest baseline cognitive score, respectively. Mid Boomers made up the largest proportion of the two middle groups (Group 2, 36.3% and Group 3, 36.2%) and the smallest proportion of the group with the highest baseline cognitive score (Group 4, 27.9%). Early Boomers comprised similar proportions of each trajectory group, ranging from 31.1% to 32.4%.

Other baseline characteristics also differed significantly across groups. Individuals in trajectory groups with lower baseline cognitive scores (Groups 1, 2, and 3) were older at baseline compared to those in the group with the highest baseline score (Group 4), more likely to be men, and more likely to identify as Black non-Latino, Latino, or of another race/ethnicity. Individuals in trajectory groups with lower baseline cognitive scores also had lower levels of educational attainment and lower net worth compared to the group with the highest cognitive baseline. Rates of hypertension, stroke, diabetes, cardiac disease, and obesity were also higher among individuals in trajectory groups with lower baseline cognitive scores.

Consistent with these descriptive findings, in unadjusted multinomial logistic regression analyses in which the group with the highest cognitive baseline was the reference, Mid Boomers had a significantly higher relative risk of being in the two middle trajectory groups compared to War Babies (relative risk [RR] for Group 2 membership, 1.58, 95% CI, 1.19–2.11; RR for Group 3 membership, 1.58, 95% CI, 1.28–1.95; S4 Table). Adjusted analyses were also consistent with descriptive findings: Early Boomers had a significantly higher risk of being in the three groups with the lowest cognitive baselines compared to War Babies (RR for Group 1, 1.47, 95% CI, 1.08–2.00; RR for Group 2, 1.35, 95% CI, 1.08–1.69;

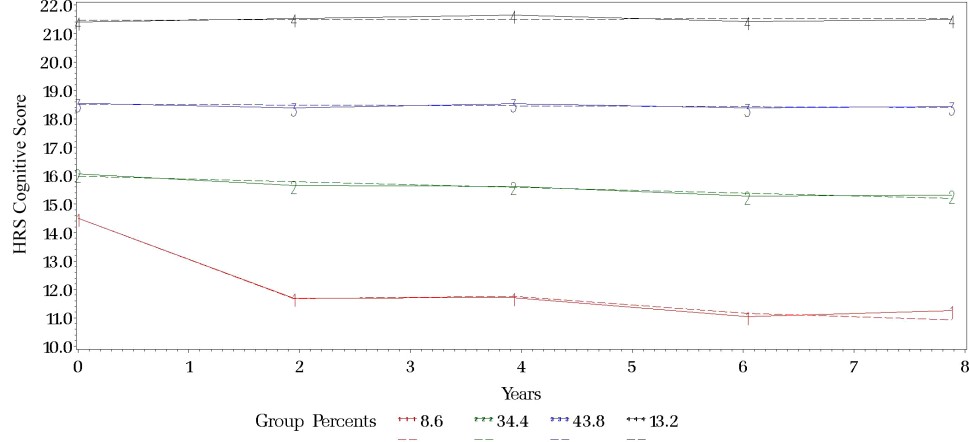

**Fig 3. Group-Based Trajectory Model for Cognitive Function Measured by HRS Cognitive Score.** Note. HRS = Health and Retirement Study. The figure shows the results of the best group-based trajectory model, identified using Akaike information criterion values. Analyses incorporated survey weights, strata, and clusters to account for the complex HRS survey design. The four groups differ primarily by baseline HRS cognitive score, with approximately linear trajectories of cognitive decline.

RR for Group 3, 1.28, 95% CI, 1.05–1.56), while Mid Boomers had a significantly higher risk of being in the two middle groups (RR for Group 2, 1.55, 95% CI, 1.18–2.04; RR for Group 3, 1.60, 95% CI, 1.31–1.97; S5 Table).

Among demographic characteristics in the adjusted multinomial logistic regression analyses, older (versus younger) individuals had a significantly higher risk of being in the group with the lowest cognitive baseline versus that with the highest cognitive baseline (S5 Table). Men (compared to women) had a higher risk of being in groups with lower cognitive baselines versus the group with the highest cognitive baseline, as did individuals who identified as Black non-Latino, Latino, or of another race/ethnicity versus those who identified as White. Those with lower (versus higher) educational attainment also had a higher risk of being in groups with lower cognitive baselines, as did those with net worth in the lowest three quartiles and those with hypertension. Of these statistically significant relative risks, the strongest relative risks (>3) were among those with lower educational attainment, those identifying as Black non-Latino, Latino, or of another race/ethnicity, and those with lower net worth. Individuals with a history of stroke, diabetes, cardiac disease, and obesity did not have a significantly higher risk of being in groups with lower cognitive baselines compared to the group with the highest cognitive baseline.

**Mixed-effects models.** In mixed-effects models examining the relationship between birth cohort and linear cognitive decline over time, unadjusted and adjusted analyses produced consistent findings (unadjusted, S6 Table; adjusted, Table 2). Mid Boomers had the lowest baseline cognitive scores (intercepts), while Early Boomers' scores were intermediate between those of Mid Boomers and War Babies (S6 Table, Table 2). However, Mid Boomers had a lower rate of cognitive decline (slope) compared to War Babies. Rates of cognitive decline did not differ significantly between Early Boomers and War Babies. Interactions between birth cohort and race/ethnicity, birth cohort and education, and birth cohort and net worth were not significant. Sensitivity analyses excluding proxies showed a similar pattern of results; baseline cognitive scores were slightly higher and slopes of decline were lower compared to the primary analyses, consistent with the more prevalent use of proxies among respondents with cognitive impairment. The number of CIND cases decreased from 388 to 332 after excluding proxies, and several unadjusted and adjusted hazard ratios were no longer statistically significant (analyses not shown).

In adjusted mixed-effects models, several other baseline characteristics were also associated with having a lower baseline cognitive score, including older age; being a man; identifying as Black non-Latino, Latino, or of another race/ethnicity versus White; having educational attainment less than college or higher; being in the lower two quartiles of net worth; and having hypertension or cardiac disease (Table 2). Of these characteristics, educational attainment was most strongly associated with baseline HRS cognitive score, followed by race/ethnicity and net worth. Characteristics associated with a steeper slope of cognitive decline included having less than a high school education (versus college or higher); being in the lower two quartiles of net worth; and having diabetes. Of these characteristics, having less than a high school education was most strongly associated with cognitive decline. Individuals who were overweight had a lower rate of cognitive decline compared to those with a BMI in the normal range.

### Incidence and risk of persistent CIND

The weighted median follow-up was 7.9 years (IQR, 7.5, 8.2) for War Babies, 7.9 years (IQR, 7.6, 8.2) for Early Baby Boomers, and 7.4 years (IQR, 6.1, 8.0) for Mid Baby Boomers. Over the follow-up period, 95 War Babies developed persistent CIND, with an incidence rate of 6.45 per 1,000 person-years. Among Early Baby Boomers, 115 developed persistent CIND (5.30 per 1,000 person-years), and among Mid Baby Boomers, 178 developed persistent CIND (4.86 per 1,000 person-years). Seventy-six War Babies died over follow-up, 114 Early Baby Boomers died, and 104 Mid Baby Boomers died.

The Kaplan-Meier curves for persistent CIND by birth cohort, both overall and stratified by race/ethnicity, showed complex patterns, including cross-over of the curves, indicating a violation of the proportional hazards assumption (Fig 4 and S1 Fig).

**Table 2. Association Between Baseline Characteristics and Intercept and Slope of HRS Cognitive Score Change (N = 7,717).**

| | Intercept, HRS Cognitive Score Change (95% CI) | Slope, HRS Cognitive Score Change (95% CI) |
|---|---|---|
| Birth Cohort | Adjusted | Adjusted |
| War Babies | Reference | Reference |
| Early Baby Boomers | **−0.316 (−0.480, −0.151)** | 0.004 (−0.025, 0.033) |
| Mid Baby Boomers | **−0.691 (−0.854, −0.527)** | **0.122 (0.092, 0.152)** |
| Age | **−0.061 (−0.099, −0.024)** | **−0.008 (−0.015, −0.002)** |
| Gender | | |
| Man | **−0.423 (−0.556, −0.289)** | **−0.036 (−0.061, −0.012)** |
| Woman | Reference | Reference |
| Race/ethnicity | | |
| White non-Latino | Reference | Reference |
| Black non-Latino | **−1.499 (−1.705, −1.293)** | −0.012 (−0.051, 0.028) |
| Other non-Latino | **−1.260 (−1.592, −0.922)** | 0.007 (−0.056, 0.070) |
| Latino | **−0.916 (−1.154, −0.678)** | −0.013 (−0.059, 0.032) |
| Educational attainment | | |
| Less than high school | **−2.445 (−2.6753, −2.2144)** | **−0.118 (−0.161, −0.076)** |
| High school or GED | **−1.734 (−1.914, −1.553)** | −0.031 (−0.064, 0.002) |
| Some college | **−1.121 (−1.293, −0.950)** | −0.002 (−0.033, 0.029) |
| College or higher | Reference | Reference |
| Net worth, quartile, $ | | |
| ≤ $17,527 | **−0.750 (−0.953, −0.546)** | **−0.054 (−0.091, −0.016)** |
| > $17,527, ≤ $71,657 | **−0.386 (−0.580, −0.192)** | **−0.038 (−0.073, −0.003)** |
| > $71,657, ≤ $188,325 | −0.083 (−0.272, 0.105) | −0.031 (−0.065, 0.003) |
| > 188,325 | Reference | Reference |
| Chronic medical conditions | | |
| Hypertension | **−0.275 (−0.424, −0.125)** | −0.004 (−0.031, 0.024) |
| Stroke | −0.281 (−0.749, 0.187) | −0.05 (−0.137, 0.037) |
| Diabetes | −0.116 (−0.338, 0.106) | **−0.045 (−0.087, −0.003)** |
| Cardiac disease | **−0.275 (−0.515, −0.035)** | 0.008 (−0.035, 0.053) |
| Body mass index | | |
| < 18.5 | −0.330 (−0.918, 0.258) | 0.099 (−0.015, 0.212) |
| 18.5-24.9 | Reference | Reference |
| 25-29.9 | 0.141 (−0.026, 0.308) | **0.032 (0.002, 0.062)** |
| ≥ 30 | 0.082 (−0.100, 0.263) | 0.021 (−0.013, 0.054) |

*Note.* CI = confidence interval; HRS = Health and Retirement Study. HRS score change calculated using mixed-effects models. The mixed-effects model included 7,717 participants, which excluded 135 individuals from the full analytic sample of 7,852, due to missing data for variables required for model estimation. Statistically significant differences in HRS scores are in bold.

In adjusted Cox regression models for persistent CIND, the interactions between race/ethnicity and birth cohort were statistically significant (p < .001 and p = .002, respectively). These interactions also violated the proportional hazards assumption. Consequently, hazard ratios (HRs) for persistent CIND are reported separately for birth cohorts by race/ethnicity and by time period (i.e., ≤ 5 years and >5 years of follow-up; persistent CIND, S7 Table, Table 3).

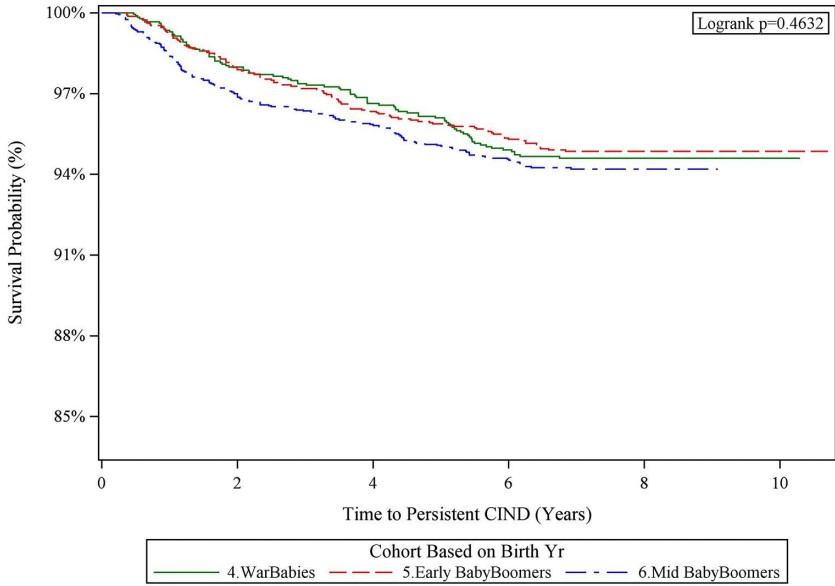

**Fig 4. Kaplan Meier Curves for Persistent CIND by Birth Cohort.** Note. CIND = Cognitive impairment no dementia. HRS = Health and Retirement Study. The figure shows the incidence of persistent CIND across the three birth cohorts. Cumulative incidences were determined using Kaplan-Meier curves. Analyses incorporated survey weights, strata, and clusters to account for the complex HRS survey design.

Among White non-Latino participants, Early and Mid Boomers were more likely to develop persistent CIND during the first half of the follow-up period (≤5 years), with HRs of 4.38 (95% CI, 2.62–7.33) and 2.68 (95% CI [1.67–4.29]), respectively, compared to War Babies (Table 3). However, this trend reversed in the second half of the follow-up period (>5 years), with HRs of 0.08 (95% CI [0.02–0.23]) and 0.05 (95% CI [0.01–0.21]), respectively. Among Latino participants, the only significant hazard ratio was for Mid Boomers compared to War Babies during the first follow-up period, with a lower hazard of CIND (HR 0.45, 95% CI [0.20–1.00]). Results from the Fine and Gray model were consistent with these findings (results not shown).

## Discussion

In this nationally representative study, we found that cognitive trajectories and incidence of CIND among middle-aged adults differed across birth cohorts. In mixed effects models, Mid Boomers had lower baseline cognitive scores compared to those born in earlier birth cohorts but a slower rate of cognitive decline. Hazards of persistent CIND were significantly higher among White non-Latino Early and Mid Boomers versus War Babies in the first half of follow-up, but lower in the second half. Among Latino participants, Mid Boomers had a lower hazard of persistent CIND compared to War Babies in the first half of follow-up. In both trajectory and incidence analyses, individuals with lower educational attainment were at highest risk for having poorer cognitive outcomes, followed by members of racial/ethnic minority groups.

Our findings complement and extend those of prior studies. Most previous studies of cognitive trajectories in middle age have combined middle-aged and older individuals or have not examined differences in trajectories across birth cohorts [50–53]. A recent study used data from four large US population-based longitudinal studies, including HRS, to examine predicted age trajectories in cognitive function across the life course, and found significant variations by 10-year birth cohorts [54]. On average, more recent birth cohorts tended to perform better on memory tests and to have slower declines in memory performance compared to earlier cohorts, possibly due to increases in educational attainment [2,45].

**Table 3. Association between Birth Cohort and Risk of Persistent CIND, Stratified by Race/Ethnicity (N = 7,331).**

| Birth Cohort, Stratified by Race/Ethnicity | Persistent CIND, 0-5 years Hazard Ratio (95% CI) | Persistent CIND, >5 years Hazard Ratio (95% CI) |
|---|---|---|
| | Adjusted | Adjusted |
| White non-Latino | | |
| War Babies | Reference | Reference |
| Early Baby Boomers | **4.38 (2.62, 7.33)** | **0.08 (0.02, 0.23)** |
| Mid Baby Boomers | **2.68 (1.67, 4.29)** | **0.05 (0.01, 0.21)** |
| Black non-Latino | | |
| War Babies | Reference | Reference |
| Early Baby Boomers | 1.02 (0.49, 2.12) | 4.50 (0.56, 36.00) |
| Mid Baby Boomers | 1.08 (0.58, 2.02) | 1.83 (0.23, 14.33) |
| Other non-Latino | | |
| War Babies | Reference | Reference |
| Early Baby Boomers | 1.17 (0.25, 5.41) | Not estimable |
| Mid Baby Boomers | 1.09 (0.27, 4.50) | Not estimable |
| Latino | | |
| War Babies | Reference | Reference |
| Early Baby Boomers | 1.01 (0.55, 1.87) | 0.20 (0.04, 1.09) |
| Mid Baby Boomers | **0.45 (0.20, 1.00)** | 0.40 (0.11, 1.40) |

*Note.* CIND = Cognitive impairment no dementia. CI = confidence interval. The mixed-effects model included 7,331 participants, which excluded 521 individuals from the full analytic sample of 7,852, due to missing data for variables required for model estimation. Statistically significant hazard ratios are in bold. Hazard ratios for the other non-Latino group are not estimable because there was no event in the reference group.

We similarly found that rates of cognitive decline were slower in later compared to earlier cohorts, which may reflect generally higher educational attainment in later cohorts. However, we found that baseline cognitive scores were lower among those born in later cohorts and that Mid and Early Boomers were also less likely to be members of the most favorable cognitive trajectory group compared to War Babies.

The conflicting findings of the current and prior study may reflect differences in birth cohort categorization: we examined 5-year birth cohorts, which distinguish between those born in the early versus middle Baby Boom (1948–1953 vs 1954–1959), while 10-year cohorts combine those individuals. We found key differences in baseline socioeconomic status and health status between 5-year cohorts, with Mid Boomers having lower educational attainment and net worth and a higher prevalence of obesity and several chronic medical conditions compared to those born in earlier cohorts. These findings are consistent with the pathways described in our conceptual model, which shows how upstream SES may "get under the skin" by influencing downstream exposures in the social and physical environment, ultimately leading to a higher prevalence of chronic health conditions that are important risk factors for AD/ADRD. These findings are also consistent with prior studies showing that those born in the mid to later part of the Baby Boom experience worse health outcomes than those born earlier in the Baby Boom, possibly due to an "Easterlin cohort effect" [55–57]. Similar to the conceptual model, this effect posits that due to the large generation size of Baby Boomers, those born later faced increased competition in the job market and lower socioeconomic status at adulthood, in turn leading to increased stress and poorer health outcomes across the life course. The finding that more recent birth cohorts had slower rates of cognitive decline despite a higher prevalence of chronic conditions suggests that their higher levels of education may have buffered

the rate of decline from a lower cognitive baseline, possibly combined with increased availability of medical treatments and technologies that can address conditions such as diabetes, hypertension, and cardiac disease [45,58].

Analyses of hazards of persistent CIND showed complex patterns. Early and Mid Boomers who identified as White non-Latino – the largest racial/ethnic group in the HRS cohort – had a significantly higher hazard of persistent CIND in years 0–5 compared to War Babies, while the hazard was significantly lower in these cohorts at >5 years. These findings suggest that the Early and Mid Boomer cohorts are comprised of subgroups of individuals at varying risk of developing CIND, with higher risk individuals developing earlier onset of CIND, leaving a lower risk group at >5 years. The group-based trajectory analyses support this explanation, showing that compared to War Babies, both Early and Mid Boomers were more likely to be members of trajectory groups with lower baseline cognitive scores compared to the group with the highest baseline scores. However, Early and Mid Boomers also both comprise nearly a third of the group with the highest baseline cognitive score and thus a potentially lower risk of incident CIND. The findings of the mixed effects models are also consistent with those of the Cox regression models, with Mid Boomers having lower baseline cognitive scores compared to War Babies.

Hazards of persistent CIND did not differ significantly across birth cohorts among individuals identifying as Black non-Latino or of another race/ethnicity, which may reflect relatively limited sample sizes among these groups by birth cohort. However, Mid Boomers identifying as Latino had a lower incidence of persistent CIND compared to War Babies during the first half of follow-up, possibly also due to higher levels of education in this later age cohort and increased availability of treatments to address risk factors for CIND [45,58].

Taken together, our findings suggest how the interplay of risk factors across birth cohorts may contribute to differences in cognitive outcomes. Our findings and those of prior studies support the key importance of early life education as well as cognitive stimulation across the life course in developing "cognitive reserve" that buffers against mid- and later life cognitive decline [2]. In the current study, lower educational attainment was the strongest risk factor for cognitive decline across analyses of both cognitive trajectories and incidence of CIND. Race/ethnicity was also strongly associated with cognitive outcomes; this association likely reflects the influence of factors not accounted for in our models, which may include educational quality and exposure to childhood adversity and aspects of the physical and social environment. Our conceptual model shows potential pathways, from SES, to environmental exposures, to chronic health conditions, through which such exposures may influence downstream cognitive outcomes. Of note, although individuals identifying as Black non-Latino, Latino, and of another race/ethnicity had lower baseline cognitive scores in mixed effects models compared to individuals identifying as White, rates of cognitive decline were similar across racial/ethnic groups, suggesting that disparities in cognitive outcomes may have their roots in earlier life experiences and exposures, consistent with the findings of prior work [59,60]. These findings suggest the importance of ongoing research to elucidate the complex pathways through which SES may influence cognitive outcomes in middle age via exposures in the social and physical environment. Understanding these pathways can help support policy efforts to bolster educational attainment and also inform the development of interventions to reduce the impact of low SES on health outcomes. These findings also underline the importance for policy makers of supporting and promoting interventions to increase educational attainment and educational quality.

This study has several limitations. The categories of CIND and dementia are based on the HRS cognitive battery, which includes a relatively limited set of measures of memory, working memory, and processing speed. However, these measures have been shown to be valid cognitive tests for use in mid-life [39–41], and a time-split validation study showed 91% concordance in the training dataset and 92% in the validation dataset for CIND and dementia diagnosis for the diagnostic algorithm compared to the ADAMS clinical evaluation for both self- and proxy-respondents [43]. Of note, this validation study did not include 1998 data, in which proxy measures of cognition were based on a 9-point rather than 11-point score. The percentage of HRS participants with proxy respondents was significantly lower at baseline and over follow-up in the Mid Boomer cohort compared to the War Babies and Early Boomers. This difference likely reflects changes in HRS procedures that were implemented in 2006, when HRS increased the proportion of face-to-face interviews administered

in participants' homes and reduced the proportion completed via telephone. This change may have led to an increase in self-interviews among cohorts recruited after 2006. It is possible that this difference may influence calibration of measures of dementia status across cohorts. Because interviews are biennial, we were unable to observe the exact date of CIND onset. Thus, we assumed a linear cognitive function change between the dates of interview before and after CIND and interpolated the date of CIND onset. Any non-linear slope of cognitive decline that occurred between two interviews, such as a sudden drop in cognitive score (e.g., after a health event such as a stroke or a traumatic brain injury) would violate this assumption. Thus, the interpolated date of CIND onset for individuals with a sudden drop in cognitive score may not be accurately specified between the two interviews. However, among the 388 persistent CIND cases, only 27 had a stroke in the same interval, suggesting limited impact on our findings. The sample size among some birth cohort groups of racial/ethnic minorities was relatively small, which may have limited our power to detect a statistically significant association with cognitive outcomes in these groups.

This nationally representative study showed mixed trends in cognitive trajectories and incidence of CIND in middle-aged Americans by birth cohort. While Early and Mid Boomers had lower baseline cognitive scores compared to War Babies in mixed effects models, Mid Boomers also had slower rates of cognitive decline. The hazard of persistent CIND was higher among White Mid Boomers compared to War Babies in the first half of follow-up but lower in the second half, likely reflecting the presence of subgroups of individuals at varying cognitive risk. Findings suggest the importance of educational attainment as a protective factor for cognitive decline, even in the setting of higher rates of chronic conditions among Mid Boomers compared to earlier age cohorts. Our findings also show that disparities in cognitive outcomes have persisted among most racial/ethnic minority groups since the late 1990s. Taken together, these findings suggest that efforts to improve cognitive outcomes in middle age will need to focus on increasing educational attainment and addressing other risk factors in earlier life, especially among racial and ethnic minorities.

## Supporting information

**S1 Table. Cut-points on the HRS Cognitive Score Self-Respondent 27-point Scale and Proxy-Respondent Scale.** Note: HRS = Health and Retirement Study. Adapted from Langa, K. M. (2020). Langa-Weir classification of cognitive function (1995 Onward). Survey Research Center Institute for Social Research, University of Michigan, p. 5. (DOCX)

**S2 Table. Proportion of Proxy Respondents by Birth Cohort over Follow-Up.** Note. Analyses incorporate survey weights, strata, and clusters to account for the complex HRS survey design. (DOCX)

**S3 Table. Distribution of Participant Characteristics by Group-Based Trajectory.** Note. GED = General Educational Development test; SD = standard deviation. (DOCX)

**S4 Table. Association of Birth Cohort with Group-Based Trajectory, Unadjusted.** Note. CI = confidence interval. Relative risk ratios calculated using multi-nominal logistic regression models. Statistically significant risk ratios are in bold. (DOCX)

**S5 Table. Association of Birth Cohort and Other Participant Characteristics with Group-Based Trajectory, Adjusted.** Note. CI = confidence interval; GED = General Educational Development test. Relative risk ratios calculated using multi-nominal logistic regression models. Statistically significant risk ratios are in bold. (DOCX)

**S6 Table. Association Between Baseline Characteristics and Intercept and Slope of HRS Cognitive Score Change, Unadjusted.** Note. CI = confidence interval; HRS = Health and Retirement Study. HRS cognitive score change calculated using mixed-effects models. Statistically significant differences in HRS scores are in bold.
(DOCX)

**S7 Table. Association of Birth Cohort with Persistent CIND, Unadjusted.** Note. CIND = Cognitive impairment no dementia. CI = confidence interval. Statistically significant hazard ratios are in bold.
(DOCX)

**S1 Fig. Kaplan Meier Curves for Persistent CIND by Birth Cohort, Stratified by Race/Ethnicity.** Note. CIND = Cognitive impairment no dementia. HRS = Health and Retirement Study. The figure shows the incidence of persistent CIND across the three birth cohorts, stratified by race/ethnicity. Cumulative incidences were determined using Kaplan-Meier curves. Analyses incorporated survey weights, strata, and clusters to account for the complex HRS survey design.
(DOCX)

## Acknowledgments

The authors do not have any additional contributors to report.

## Author contributions

**Conceptualization:** Rebecca T. Brown, Xi Zuo, Corey T. McMillan, Shana D. Stites, Irma T. Elo, Kenneth M. Langa, Dawei Xie.

**Formal analysis:** Xi Zuo, Dawei Xie.

**Funding acquisition:** Rebecca T. Brown.

**Investigation:** Rebecca T. Brown, Xi Zuo, Corey T. McMillan, Shana D. Stites, Irma T. Elo, Kenneth M. Langa, Jason Karlawish, Dawei Xie.

**Methodology:** Rebecca T. Brown, Xi Zuo, Corey T. McMillan, Shana D. Stites, Dawei Xie.

**Project administration:** Rebecca T. Brown, Dawei Xie.

**Resources:** Rebecca T. Brown.

**Supervision:** Rebecca T. Brown, Corey T. McMillan, Dawei Xie.

**Validation:** Rebecca T. Brown, Xi Zuo, Corey T. McMillan, Shana D. Stites, Irma T. Elo, Kenneth M. Langa, Jason Karlawish, Dawei Xie.

**Visualization:** Rebecca T. Brown, Xi Zuo, Shana D. Stites, Irma T. Elo, Kenneth M. Langa, Jason Karlawish, Dawei Xie.

**Writing – original draft:** Rebecca T. Brown, Dawei Xie.

**Writing – review & editing:** Rebecca T. Brown, Xi Zuo, Corey T. McMillan, Shana D. Stites, Irma T. Elo, Kenneth M. Langa, Jason Karlawish, Dawei Xie.

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
