## [Decision Letter · Decision Letter 0]

21 May 2025

Dear Dr. Brown,

Thank you for submitting your manuscript to PLOS ONE. After careful consideration, we feel that it has merit but does not fully meet PLOS ONE’s publication criteria as it currently stands. Therefore, we invite you to submit a revised version of the manuscript that addresses the points raised during the review process.

We look forward to receiving your revised manuscript.

Kind regards,

Amruta Deshpande

Academic Editor

PLOS ONE

 [This work was supported by the National Institute on Aging at the National Institutes of Health (grant number R01AG070885 to R.T.B.; grant numbers K23AG065442 and K23AG065442-03S1 to S.D.S.; P30AG072979 to C.T.M) and Penn Institute on Aging (no grant number to C.T.M.). The Health and Retirement Study is supported by the National Institute on Aging (grant number U01AG009740) and performed at the Institute for Social Research, University of Michigan.].

Additional Editor Comments:

Mayor revisión . work on comments given by reviewwers and submit the revised manuscript

Reviewers' comments:

Reviewer's Responses to Questions

**Comments to the Author**

1. Is the manuscript technically sound, and do the data support the conclusions?

Reviewer #1: Yes

Reviewer #2: No

2. Has the statistical analysis been performed appropriately and rigorously?

Reviewer #1: Yes

Reviewer #2: Yes

3. Have the authors made all data underlying the findings in their manuscript fully available?

Reviewer #1: Yes

Reviewer #2: Yes

4. Is the manuscript presented in an intelligible fashion and written in standard English?

Reviewer #1: Yes

Reviewer #2: Yes

Reviewer #1: Dear Authors

Your article sheds light to a valid concept and work is well done. I believe you can consider these extra changes to make your excellent work perfect:

1. Proxy Respondent Calibration: Consider a sensitivity analysis excluding proxy respondents to assess the robustness of cohort comparisons, given changes in HRS data collection procedures after 2006.

2. CIND Onset Estimation: Clarify or address the potential misclassification of CIND onset due to the assumption of linear cognitive decline between biennial assessments. A subgroup analysis excluding participants with stroke could help.

3. Supplementary Material Integration: Summarize key findings from supplementary tables (e.g., adjusted trajectory group membership) in the main text for better accessibility.

4. Sample Size Caveats: Explicitly acknowledge the limited statistical power in racial/ethnic minority subgroup analyses when interpreting null results.

Reviewer #2: This manuscript addresses an important topic by examining cognitive outcomes using data from the Health and Retirement Study (HRS). While the use of a large, nationally representative dataset and the authors’ effort to consider weights and competing risks, such as mortality, are commendable, I have significant concerns regarding the validity of the core analytic approach and the lack of clarity in model specification. My comments below outline specific concerns that would require significant clarification and methodological refinement for the study to meet the standards of rigor and interpretability expected for publication.

1. The introduction would benefit from revision to provide a clearer and more compelling rationale for the study. Specifically, it is not well explained why differences in cognitive outcomes across individual birth cohorts should be expected (and in which direction). A stronger theoretical or empirical justification for examining cohort effects is needed to frame the research question. Additionally, the authors refer to examining “cognitive trajectories and incidence of cognitive decline in the form of “cognitive impairment no dementia” (CIND, often a precursor of dementia),” but no incidence rates appear to be reported in the manuscript. If incidence was not estimated, this terminology should be removed or revised to more accurately reflect the outcomes analyzed (e.g. the relative hazard of CIND).

2. My primary concern is the application of the cognitive status algorithm to a relatively young cohort (ages 50–54). According to Petersen et al., the prevalence of mild cognitive impairment among individuals aged 60–64 is approximately 6.7%. Therefore, the finding that approximately 20% of participants aged 50–54 (1,935 out of 10,896) were classified as having CIND or dementia at baseline appears highly implausible and suggests potential misclassification. This raises substantial concerns about the validity of the algorithm when applied to younger age groups. The algorithm was originally developed and validated using data from ADAMS, which seems to have included only individuals aged 70 and older. I recommend the authors provide strong justification for the algorithm’s use in this age group and whether such a high number is realistic or consider alternative approaches more appropriate for middle-aged adults.

Petersen RC, Lopez O, Armstrong MJ, Getchius TSD, Ganguli M, Gloss D, Gronseth GS, Marson D, Pringsheim T, Day GS, Sager M, Stevens J, Rae-Grant A. Practice guideline update summary: Mild cognitive impairment: Report of the Guideline Development, Dissemination, and Implementation Subcommittee of the American Academy of Neurology. Neurology. 2018 Jan 16;90(3):126-135. doi: 10.1212/WNL.0000000000004826. Epub 2017 Dec 27. PMID: 29282327; PMCID: PMC5772157.

3. It is not clear whether the “key covariates” included in the analysis are intended to be treated as confounders, effect modifiers, or whether included for other reasons. Could some of these be mediators? Conceptually, variables included in the model to control for bias in the exposure-outcome relationship should be referred to as confounders. The generic label “covariate” lacks clarity. Moreover, presenting and interpreting effect estimates for covariates included to adjust for confounding is discouraged. Specifically, paragraph 4 in the Group-based trajectory models section and paragraph 2 (and Table 2) in the Linear-mixed effects models section display and interpret effect estimates for confounding variables. For guidance, please see: Westreich et al. for more details: Westreich D, Greenland S. The table 2 fallacy: presenting and interpreting confounder and modifier coefficients. Am J Epidemiol. 2013 Feb 15;177(4):292-8. doi: 10.1093/aje/kws412. Epub 2013 Jan 30. PMID: 23371353; PMCID: PMC3626058.

4. In addition, the authors might consider excluding the p-values for differences in Table 1. Please see: Hayes-Larson E, Kezios KL, Mooney SJ, Lovasi G. Who is in this study, anyway? Guidelines for a useful Table 1. J Clin Epidemiol. 2019 Oct;114:125-132. doi: 10.1016/j.jclinepi.2019.06.011. Epub 2019 Jun 20. PMID: 31229583; PMCID: PMC6773463.

5. The Cox model needs to be specified in more detail. It is unclear what was used as the time axis - age or follow-up time? Was the model left-truncated?

6. Similarly, the linear mixed-effects model needs to be described in more detail. It is not specified whether time was modelled as age or as years since baseline. If time was modeled as years since baseline, it is unclear whether there was an overall cognitive decline observed in the cohort. Further, it appears that interaction terms between time and each covariate were included (in addition to interactions between birth cohort and selected covariates listed in methods). If so, this likely increases model complexity. A more parsimonious model may be preferable.

7. What was the proportion of missing data in the covariates? And how were missing data handled? It would be helpful to have a flow diagram illustrating how the final analytic sample was derived from the whole HRS cohort.

8. Were any participants categorized as having dementia during follow-up? If so, how were they treated? What was the number of CIND events during the follow-up period?

9. I appreciate that the authors considered the competing risk of death in their analyses. The consistency of the results across these models adds confidence to the robustness of the findings. However, would it be possible to include more details on mortality patterns in the descriptive statistics?

10. What was the mean (or median) length of follow-up.

11. Please include number of participants (N) in Tables 2 and 3 and the corresponding supplemental tables.

12. The title suggests a follow-up period of 20 years, which may be misleading. The maximum follow-up within any single birth cohort was 8 years. I understand that the 20-year figure refers to the total span of HRS data across all cohorts, but I would recommend revising the title to avoid potential confusion.

**Do you want your identity to be public for this peer review?** For information about this choice, including consent withdrawal, please see our Privacy Policy

Reviewer #1: **Yes: ** Dr. Babak Najand

Reviewer #2: No

---

## [Author Response · Author response to Decision Letter 1]

3 Jul 2025

Please see attached "Response to Reviewers"

---

## [Decision Letter · Decision Letter 1]

29 Aug 2025

Dear Dr. Brown,

Thank you for submitting your manuscript to PLOS ONE. After careful consideration, we feel that it has merit but does not fully meet PLOS ONE’s publication criteria as it currently stands. Therefore, we invite you to submit a revised version of the manuscript that addresses the points raised during the review process.

We look forward to receiving your revised manuscript.

Kind regards,

Amruta Deshpande

Academic Editor

PLOS ONE

Journal Requirements:

Additional Editor Comments:

All the best

Reviewers' comments:

Reviewer's Responses to Questions

**Comments to the Author**

Reviewer #3: All comments have been addressed

2. Is the manuscript technically sound, and do the data support the conclusions?

Reviewer #3: Yes

3. Has the statistical analysis been performed appropriately and rigorously?

Reviewer #3: Yes

4. Have the authors made all data underlying the findings in their manuscript fully available?

Reviewer #3: Yes

5. Is the manuscript presented in an intelligible fashion and written in standard English?

Reviewer #3: Yes

Reviewer #3: Dear Author,

The research work certainly brings out, output with rigorous approach, however, to add more strength to the discussion work following points are suggested:

1. Bringing academic knowledge discussion part on the conceptual model deeply by discussing each causal pathway relationship and its implication to medical and healthcare fraternity in macro view to both research field and medical fraternity policy makers and experts for their utility.

2. If we can propose benefits of this model to the stakeholders of medical and healthcare field.

**Do you want your identity to be public for this peer review?** For information about this choice, including consent withdrawal, please see our Privacy Policy

Reviewer #3: No

---

## [Author Response · Author response to Decision Letter 2]

4 Sep 2025

September 3, 2025

Amruta Deshpande, Academic Editor

PLOS One

Dear Dr. Deshpande,

We thank the Editor and Reviewers for their comments and appreciate the opportunity to address the comments and revise our manuscript. Below, please find item-by-item responses to the comments, which are included verbatim. All page, paragraph, and line numbers refer to locations in the revised manuscript and online-only supplement. We have uploaded a clean copy of the revised manuscript as well as a copy of the manuscript with tracked changes.

a. Response: We have included the requested documents in our resubmission.

Responses to Reviewer 3:

1. Dear Author,

The research work certainly brings out, output with rigorous approach, however, to add more strength to the discussion work following points are suggested:

1. Bringing academic knowledge discussion part on the conceptual model deeply by discussing each causal pathway relationship and its implication to medical and healthcare fraternity in macro view to both research field and medical fraternity policy makers and experts for their utility.

a. Response: We thank the reviewer for this comment and opportunity to further discuss the conceptual model and its implications, and we have edited the discussion accordingly. Of note, the current study focuses on understanding how cognitive outcomes vary across birth cohorts, rather than on examining mechanisms of cognitive decline or identifying novel risk factors (see Methods, page 11, paragraph 1). For this reason, the study includes variables that are established risk factors for cognitive decline in the proposed conceptual model and have been included in prior studies of trends in cognitive outcomes over time. These include sociodemographic characteristics, measures of socioeconomic status, and measures of health status. For this reason, the discussion does not consider every pathway in the conceptual model, as not all of these are relevant to the current manuscript. However, we have added additional discussion of the relevant pathways in the conceptual model to the discussion, as follows:

b. Page 24, paragraph 1: “The conflicting findings of the current and prior study may reflect differences in birth cohort categorization: we examined 5-year birth cohorts, which distinguish between those born in the early versus middle Baby Boom (1948-1953 vs 1954-1959), while 10-year cohorts combine those individuals. We found key differences in baseline socioeconomic status and health status between 5-year cohorts, with Mid Boomers having lower educational attainment and net worth and a higher prevalence of obesity and several chronic medical conditions compared to those born in earlier cohorts. These findings are consistent with the pathways described in our conceptual model, which shows how upstream SES may “get under the skin” by influencing downstream exposures in the social and physical environment, ultimately leading to a higher prevalence of chronic health conditions that are important risk factors for AD/ADRD. These findings are also consistent with prior studies showing that those born in the mid to later part of the Baby Boom experience worse health outcomes than those born earlier in the Baby Boom, possibly due to an “Easterlin cohort effect” (55-57) Similar to the conceptual model, this effect posits that due to the large generation size of Baby Boomers, those born later faced increased competition in the job market and lower socioeconomic status at adulthood, in turn leading to increased stress and poorer health outcomes across the life course. The finding that more recent birth cohorts had slower rates of cognitive decline despite a higher prevalence of chronic conditions suggests that their higher levels of education may have buffered the rate of decline from a lower cognitive baseline, possibly combined with increased availability of medical treatments and technologies that can address conditions such as diabetes, hypertension, and cardiac disease (45, 58).”

2. If we can propose benefits of this model to the stakeholders of medical and healthcare field.

a. Response: We agree that further discussion of the benefits of the model for the medical and health care fields would strengthen the discussion. We have highlighted key benefits of using a conceptually informed approach, as follows:

b. Page 26, paragraph 1: “Taken together, our findings suggest how the interplay of risk factors across birth cohorts may contribute to differences in cognitive outcomes. Our findings and those of prior studies support the key importance of early life education as well as cognitive stimulation across the life course in developing “cognitive reserve” that buffers against mid- and later life cognitive decline (2). In the current study, lower educational attainment was the strongest risk factor for cognitive decline across analyses of both cognitive trajectories and incidence of CIND. Race/ethnicity was also strongly associated with cognitive outcomes; this association likely reflects the influence of factors not accounted for in our models, which may include educational quality and exposure to childhood adversity and aspects of the physical and social environment. Our conceptual model shows the pathways, from SES, to environmental exposures, to chronic health conditions, through which such exposures may influence downstream cognitive outcomes. Of note, although individuals identifying as Black non-Latino, Latino, and of another race/ethnicity had lower baseline cognitive scores in mixed effects models compared to individuals identifying as White, rates of cognitive decline were similar across racial/ethnic groups, suggesting that disparities in cognitive outcomes may have their roots in earlier life experiences and exposures, consistent with the findings of prior work (59, 60). These findings suggest the importance of ongoing research to elucidate the complex pathways through which SES may influence cognitive outcomes in middle age via exposures in the social and physical environment. Understanding these pathways can help support policy efforts to bolster educational attainment and also inform the development of interventions to reduce the impact of low SES on health outcomes. These findings also underline the importance for policy makers of supporting and promoting interventions to increase educational attainment and educational quality.”

We thank the Editors and Reviewers for their time and efforts to improve our manuscript.

Sincerely,

Rebecca Brown, MD, MPH

Associate Professor of Medicine, Division of Geriatric Medicine

Perelman School of Medicine of the University of Pennsylvania

---

## [Decision Letter · Decision Letter 2]

23 Nov 2025

Trends in Cognitive Outcomes in Middle-Aged Americans Across Three Birth Cohorts

PONE-D-25-16575R2

Dear Dr. Brown,

We’re pleased to inform you that your manuscript has been judged scientifically suitable for publication and will be formally accepted for publication once it meets all outstanding technical requirements.

Kind regards,

Amruta Deshpande

Academic Editor

PLOS ONE

Additional Editor Comments (optional):

congratulations

Reviewers' comments:

Reviewer's Responses to Questions

**Comments to the Author**

Reviewer #3: All comments have been addressed

2. Is the manuscript technically sound, and do the data support the conclusions?

Reviewer #3: Yes

3. Has the statistical analysis been performed appropriately and rigorously?

Reviewer #3: Yes

4. Have the authors made all data underlying the findings in their manuscript fully available?

Reviewer #3: Yes

5. Is the manuscript presented in an intelligible fashion and written in standard English?

Reviewer #3: Yes

Reviewer #3: Revisions made as per the earlier reviewer comments.

The research work certainly brings out, output with rigorous approach, however, to

add more strength to the discussion work following points are suggested:

1. Bringing academic knowledge discussion part on the conceptual model deeply by

discussing each causal pathway relationship and its implication to medical and

healthcare fraternity in macro view to both research field and medical fraternity

policy makers and experts for their utility.

Second suggestion has been also implemented of Industry implications.

So it is implied that the

**Do you want your identity to be public for this peer review?** For information about this choice, including consent withdrawal, please see our Privacy Policy

Reviewer #3: **Yes: ** Dipanjay Bhalerao

---

## [Editor Report · Acceptance letter]

PONE-D-25-16575R2

PLOS ONE

Dear Dr. Brown,

I'm pleased to inform you that your manuscript has been deemed suitable for publication in PLOS ONE. Congratulations! Your manuscript is now being handed over to our production team.

Kind regards,

on behalf of

Dr. Amruta Deshpande

Academic Editor

PLOS ONE